# Sickle Cell Disease Genomics of Africa (SickleGenAfrica) Network: ethical framework and initial qualitative findings from community engagement in Ghana, Nigeria and Tanzania

Kofi A Anie [1,2] Edeghonghon Olayemi,[3,4] Vivian Paintsil,[5,6] Ellis Owusu-Dabo,[7] Titilope Adenike Adeyemo,[8] Mahmoud U Sani,[9] Najibah Aliyu Galadanci,[10] Obiageli Nnodu,[11] Furahini Tluway,[12] David Nana Adjei,[13] Peter Mensah,[14] Joseph Sarfo-Antwi,[15] Henry Nwokobia,[16] Awwal Gambo,[17] Adebola Benjamin,[18] Arafa Salim,[19] Judith A Osae-Larbi,[20] Solomon Fiifi Ofori-Acquah,[13,20] On behalf of the SickleGenAfrica Network

We endure the sad loss of HN and honour his meticulous contribution as a community liaison.

**Correspondence to**
Dr Kofi A Anie;
kofi.anie@nhs.net

## ABSTRACT

**Objectives** To provide lay information about genetics and sickle cell disease (SCD) and to identify and address ethical issues concerning the Sickle Cell Disease Genomics of Africa Network covering autonomy and research decision-making, risk of SCD complications and organ damage, returning of genomic findings, biorepository, data sharing, and healthcare provision for patients with SCD.

**Design** Focus groups using qualitative methods.

**Setting** Six cities in Ghana, Nigeria and Tanzania within communities and secondary care.

**Participants** Patients, parents/caregivers, healthcare professionals, community leaders and government healthcare representatives.

**Results** Results from 112 participants revealed similar sensitivities and aspirations around genomic research, an inclination towards autonomous decision-making for research, concerns about biobanking, anonymity in data sharing, and a preference for receiving individual genomic results. Furthermore, inadequate healthcare for patients with SCD was emphasised.

**Conclusions** Our findings revealed the eagerness of patients and parents/caregivers to participate in genomics research in Africa, with advice from community leaders and reassurance from health professionals and policy-makers, despite their apprehensions regarding healthcare systems.

## INTRODUCTION

Sickle cell disease (SCD) is designated by the WHO[1] and the United Nations General Assembly[2] as a major global health burden. Over 300 000 babies are born worldwide annually with SCD, with ~75% in sub-Saharan Africa.[3] Typical SCD clinical syndromes include severe haemolytic anaemia, and the consequences of vaso-occlusion indicated

## Strengths and limitations of this study

► Community engagement is a critical introductory and ethical step in human genomics research.
► Qualitative research methodology was used to conduct community engagement focus groups at six study sites.
► Participants represented diverse local communities within three countries in West and East Africa.
► There was a potential bias in the selection of participants by invitation to focus groups.
► The targeted number of participants was not achieved at all study sites.

in acute pain, acute chest syndrome, stroke and organ damage. Deficient early diagnosis and prophylaxis against bacteraemia causes early childhood mortality in Africa.[4] End-stage organ damage causes SCD mortality in adults.[5]

The Human Heredity and Health in Africa (H3Africa) initiative offers opportunities to advance genomic research in Africa.[6] The Sickle Cell Disease Genomics of Africa (SickleGenAfrica) Network within H3Africa is a genomics project across six African cities. SickleGenAfrica postulates that genetic variation influences the body's defence against haemolysis with the risk of end-stage organ damage.

Salient ethical considerations in genomic research include biological sample management, biorepository, data storage, sharing of de-identified information and potential informational harm posed by return of results to research participants. SickleGenAfrica is

**Table 1** Ethics approval and participating institutions of the SickleGenAfrica Network

| Country | City | Institution |
|---|---|---|
| Ghana | Accra | University of Ghana |
| | Kumasi | Kwame Nkrumah University of Science and Technology and Komfo Anokye Teaching Hospital |
| Nigeria | Lagos | Lagos University Teaching Hospital |
| | Kano | Aminu Kano Teaching Hospital |
| | Abuja | University of Abuja |
| Tanzania | Dar es Salaam | Muhimbili University of Health and Allied Sciences |

investigating three fundamental ethical issues, namely autonomy, best interest and duty of care through community engagement. Autonomy is central to informed consent; we are exploring how research participants in Ghana, Nigeria and Tanzania will act in accordance with their societal norms (value systems) regarding autonomy in informed consent provision. Participants will be considered autonomous if they provide informed consent in accordance with their own individual values, integrity and privacy and without coercion from significant others. The best interest of participants will be examined in terms of preferences necessary to mitigate potential informational harm, including what, when and how genomic results will be released to participants within H3Africa guidelines. Duty of care ensures non-maleficence and health of patients. SickleGenAfrica may disclose information to participants with the prospect of personalised medicine in future. Nonetheless, individuals' healthcare needs and well-being will be considered, including psychosocial support available.

Community engagement is a critical preparatory and ethical step in human genomic research especially within African vulnerable populations. The primary objectives of our community engagement are to:

► Conduct focus groups with stakeholders over the duration of the project to provide lay information and explore and address concerns including ethics on all aspects of SickleGenAfrica.
► Administer questionnaires on ethical issues and barriers to patient recruitment and assess participant satisfaction.

We report preliminary findings from focus groups.

## METHODS

Research ethics approvals were obtained from participating institutions (Table 1).

### Patient and public involvement

Prior to recruitment, we convened a Community Engagement Working Group of the study's lead clinicians and 'Community Liaisons' to recruit participants known to them for focus groups with an identical composition at six sites in Ghana (Accra, Kumasi), Nigeria (Abuja, Kano, Lagos) and Tanzania (Dar es Salaam). Twenty participants for each site were targeted by invitation letter, including patients, parents/caregivers, community leaders, healthcare professionals and government representatives. None of the participants refused an invitation; however, some of them did not attend without a reason (table 2). All community liaisons together with lead clinicians were invited as observers to the first focus group workshop held in Accra and are coauthors who will be involved in the dissemination of findings.

### Procedure

Three-hour focus groups were moderated by two psychologists, KAA (male) and MAA (female), both experienced

**Table 2** SickleGenAfrica year 1 focus groups: participants at each study site

| Study site/participants and target number | Accra | Kumasi | Abuja | Kano | Lagos | Dar es salaam |
|---|---|---|---|---|---|---|
| Parents/caregivers of children with SCD (4) | 4 | 4 | 3 | 4 | 3 | 4 |
| Adults with SCD (4) | 4 | 4 | 2 | 3 | 4 | 4 |
| Patient association/support group (2) | 2 | 2 | 2 | 2 | 2 | 2 |
| Community leaders (3) | 3 | 2 | 2 | 2 | 3 | 3 |
| Doctors (2) | 2 | 2 | 2 | 2 | 2 | 2 |
| Hospital nurse (1) | 1 | 1 | 1 | 1 | 1 | 1 |
| Community nurse (1) | 1 | 1 | 1 | 1 | 1 | 1 |
| Community health worker (1) | 1 | 1 | 1 | 1 | 1 | 1 |
| Government health representatives (2) | 2 | 2 | 2 | 2 | 2 | 2 |
| Total number | 20 | 19 | 16 | 18 | 19 | 20 |
| Proportion of targeted number (%) | 100 | 95 | 80 | 90 | 95 | 100 |

SCD, sickle cell disease; SickleGenAfrica, Sickle Cell Disease Genomics of Africa Network.

| Table 3 | SickleGenAfrica year 1 focus groups: statements and questions for discussion |
| --- | --- |
| **Category** | **Statements and questions** |
| Autonomy | Decisions have to be voluntary and free from coercion.<br>How would you decide about research participation?<br>▶ Personal/individual.<br>▶ Family member.<br>▶ Head of household.<br>▶ Family elder.<br>▶ Community or religious leader.<br>▶ Other. |
| Best interest: genomics and sickle cell disease | Making a best interest decision about receiving information with potential informational harm regarding: chances (i.e.risks) of sickle cell complications and organ damage in a patient, and there is no immediate treatment.<br>▶ Would you want to know the chances?<br>▶ Should the patient or caregiver be told?<br>▶ Would it be beneficial?<br>▶ Would it be harmful? |
| Biorepository | Samples will be stored in a special place called a biobank for many years.<br>Samples may be used for future research.<br>▶ Do you have any concerns? |
| Data sharing | Your information will be shared with other researchers who are part of SickleGenAfrica and others outside the group.<br>Your personal information will be removed first, for example, name.<br>▶ Do you want your information to be shared within the SickleGenAfrica group only?<br>▶ Do you want your information to be shared outside the SickleGenAfrica group?<br>▶ Do you have any concerns? |
| Duty of care | What does the government need to do for sickle cell patients?<br>Are there any best practices to adopt? |

SickleGenAfrica, Sickle Cell Disease Genomics of Africa Network.

in SCD and qualitative research, who have no prior relationships with participants. Statements and semistructured questions were generated for discussions (table 3); translators and scribes were present. Major issues covered included lay information about genes and SCD inheritance (by demonstration using coloured stringed beads), research ethics and informed consent, risk of SCD organ damage identification, sample collection, biorepository, data sharing, feedback of findings, and healthcare provision for patients with SCD. Participants were encouraged to be candid and speak in their preferred language. Their questions and concerns about SickleGenAfrica research were addressed accordingly. Proceedings were audio-recorded and notes were taken by a scribe. Inaugural focus groups were conducted over 2 weeks across all study sites within hospital or community settings. Audio recordings were transcribed and coded for themes to be reported.

## RESULTS
A total of 112 (94%) out of expected 120 participants attended the focus groups. The composition is presented in table 2. Emergent themes from coded data relating to research ethics are summarised with sample participant vignettes.

### Expectations of information for consent
Discussions about decision-making to participate in research revealed three thematic categories regarding participants' expectations of information required from researchers to solicit informed consent.

### Provision of comprehensive information
Participants emphasised the value of detailed lay information about a study, including its importance and rationale prior to consent. They indicated that patients and carers have a right to information and direction to appropriate information on the internet if available. The importance of initial communication and updates to help prevent withdrawals was also emphasised.

> We find researchers who just come in the context of an informed consent and all they throw on the participant is just, I am this, I want to do this, I will take your blood without properly letting the person know what you are going to do, so if the people are properly informed and they are well informed of what you are going to do, the benefit, the hazard and everything you will find out that getting consent will not be a problem. (FG3 Abuja)

### Type of study
An explanation of the type of study needs to be given, for example, a drug trial or an experimental study, and whether it is potentially harmful or non-invasive. Some participants expressed particular concerns about clinical trials.

I think the thing is that we suffer so much and anybody don't want to be the guinea pig… I think here in this part of the world if something goes wrong it will be difficult to rectify but the western world if something goes wrong with the clinical trial I think rectifying it is easier than here. (FG5 Lagos)

### Benefits of a study

Participants expressed the need to be told about potential benefits of any study, that is, benefits versus risks and the pros and cons of participation.

…one thing that they find very important is the benefit, they are very interested in the benefit of the study to themselves, so I believe that full disclosure is very, very important. (FG1 Accra)

### Autonomy

Autonomy was explored in terms of research participation decision-making, that is, how people would voluntarily decide to participate in a study such as SickleGenAfrica and who they tend to consult within the context of family systems and cultural settings.

### Personal decision

Personal decision-making is fairly common. Some participants believed that if you are a patient with SCD or parent/caregiver, you are also responsible for yourself or your child in key decisions about health, treatment and other choices including research. Therefore, an individual would decide and inform others such as a spouse or family members afterwards, stating that there was no reason to consult others or consent was not required from others.

…the decision is mine but because my family is close to me and know my status since I was born, I will share with them. (FG6 Dar es Salaam)

Please the men don't even come to the hospital its always the women who do, the man will not come it is you the mother who knows what you are going through, so you will decide yourself and when you go home and tell him, he will agree to it. (FG2 Kumasi)

### Family decision

Some participants identified conferring with family members regarding research as a customary process, which includes consultation with a husband or head of household (usually male) first for his agreement to consent, parental restrictions and consultation with other family members for agreement and support, that is, 'gate keeping culture'.

I am a parent, in such a research definitely I need to consult my husband before I make a move because I can't take a decision by myself without his own input; I am a patient…I am always with my parents and before I will answer any question I am supposed to let them know. (FG3 Abuja)

…nature of the family system we run here, definitely you have to discuss with your mother or your father, you know with we Ghanaians we normally ply the communal way of living… (FG1 Accra)

Conversely, the type of study could determine autonomy exercised by people with SCD.

…if it's something that in the long run, perhaps it is something that will cause harm and all that, I may have to consult with family first. (FG5 Lagos)

### Significant others' decision

A number of significant others were recognised to be involved in research participation decision-making. Focus group participants acknowledged religious and community leaders, friends, and experts such as doctors who are independent of a particular study.

I think when an individual is going for the research is good for the community leaders to know…. (FG4 Kano)

In terms of research participation, depending on the area, if it [the genomics research] is in a community you can't enter a community without the concern of the community leader. (FG4 Abuja)

…there are also some too their backing is not from the husband or from the wife, is from their religious leader…. (FG1 Accra)

…most at times it is their friends because I have quite a number of youth coming to the clinic, most of the youth, they always discuss with their friends and that is why I have been trying to encourage them to have this sickle cell club…. (FG5 Lagos)

Furthermore, consultation with significant others could depend on the type of research.

…it depends on the type of research, if you are going to give a drug or something to that child so it is important to involve the religious, the community and all other…. (FG4 Kano)

### Best interest: genomics and SCD

Genomic research and potential informational harm relating to the return of individual results about risks of SCD complications and organ damage were explained comprehensively to focus group participants. Also, they were informed that patients enrolled in SickleGenAfrica would not have direct therapeutic benefits at the time their results were reported; however, they would contribute to a 'greater good' in more precise future treatment for people with SCD.

Majority of focus group participants approved the return of individual findings; nonetheless, they emphasised that these results should be presented sensitively with some psychosocial, genetic counselling and family support. This would perhaps allow patients to make lifestyle modifications, self-manage their SCD better or

prepare for the worst, such as death including writing a will.

> …as a parent I will like the participant to know about the result to know about what is happening so that I can prepare for the future. (FG4 Kano)

> Well I think it is important that the patient is told the truth so that he/she knows how to carry himself or herself, but I believe that it depends on the presentation how you present the issue to the patient is also very important…. (FG1 Accra)

> I can be told but counselling is very important…. (FG6 Dar es Salaam)

Nonetheless, some participants were cautious about unnecessary fear or 'labelling' of patients with risks of complications, that is, stigmatisation, which may affect parental care. Some doctors also stated that they could not communicate bad results or were not ready to disclose undesirable information.

> I don't want to be told because you are doing the research and you don't have enough information for me so I don't want to be told as a parent. That one will save me as a parent in my own little way to be able to support my son or daughter confidently rather than be bias. (FG3 Abuja)

> ….I will set my house in order, I will know how to prepare myself bracing the days ahead of me, so I believe that they need to be told but the presentation must be such that it doesn't even destroy the person but it builds the person up. (FG1 Accra)

## Biorepository

The purpose of a biorepository was described before discussions commenced with a focus on acceptability of the concept of storage. Information was provided about the H3Africa Biorepository in Abuja, where SickleGenAfrica study samples will ultimately be stored. Four main categories of concerns were established.

### Quality control

Majority of participants across the study sites did not raise any specific issues regarding a biorepository.

> …it will help… if we store some blood somewhere and then in future more research will be done on it, I don't have any problem with that. (FG2 Kumasi)

Nonetheless, many participants in Nigeria expressed concerns about preservation of samples at the biorepository in Abuja owing to stability and sustainability of power supply. Although their fears were allayed by the focus group facilitators, some participants were not persuaded to change their views that are influenced by unreliable power supply in Nigeria overall.

> …the only concern that I will have about a biobank is maybe because I live in Nigeria and there is always or we don't have electricity constantly how will this sample survive so I think that is the only concern I have. (FG3 Abuja)

### Errors and accidents

There was some apprehension about human error, including the possibility of missing, misplaced or mislabelling samples, and issues during transportation such as safety and accidents. According to participants, these could mean that they may be required to provide replacement samples at study sites, which could be discouraging to patients and may also lead to unnecessary stress or anxiety.

> Perhaps a sample is mislabelled just at the point of collection, what is done or what securities are in place to ensure that there is no mislabelling, you know like we have been talking about human error it could happen and on the long run we now have this person's results being reported to another person. (FG5 Lagos)

> …and along the line through the transport of this sample the sample gets missing, and they say the patients have to go through the stress all over again of doing this procedure. Please how secure will this transport be so that they don't come back again and say sorry this sample got missing along the line and we have to take another sample again, you know that will discourage people from continuing with the project, thank you. (FG5 Lagos)

### Trust, security and sample exploitation

In addition, trust was an important issue. A small number of participants expressed the importance of samples being used ethically for intended purposes of informed willingness to consent to the future use of their samples whether alive or dead.

> …so far as I have agreed for my blood sample to be taken for the research that means I have given advance permission, so when it is time whether I am alive or dead and you say it is being used for research, you are free to do research…. (FG2 Kumasi)

However, there were concerns about the importance of samples being used ethically for intended purposes in line with informed consent. Furthermore, the credibility of biorepository workers in terms of sample misappropriation was discussed; specifically, samples should not be exploited for other reasons or financial gain by biorepository workers, and participants stated that biological samples are precious.

> …people have always been afraid of people misusing especially genetic information and apart from that so far as all the precaution of keeping – like you said it is a bank, a bank must be properly secured so that people do not get into it without authority, that's all. (FG3 Abuja)

I don't have any problem for a sample being kept for future researches but …then you watch a films where a sample that was kept in the care of lab researchers being used for something else that you have not asked them to use and another thing, how are we even sure that this person that has the code numbers will not sell us out…. (FG5 Lagos)

### Data sharing

Participants were informed that data sharing is an important component of genomic research such as SickleGenAfrica; however, this can reveal sensitive information that raises privacy and security issues. Therefore, procedures are in place to remove identifying individual information to render the data 'de-identified' or anonymous. This includes 'pseudo-anonymization' by coding and disguising the source of the data, which helps to protect individuals' information that might reveal something they prefer to keep private. Moreover, data could be shared many years after consent has been obtained, although patients could withdraw their consent at any time from whichever aspect of the study, including sample storage and data sharing.

Participants were asked to distinguish between data sharing among the SickleGenAfrica group of researchers and scientists outside this group. Overall, there were no major concerns about data sharing, with only a few who preferred to restrict sharing to SCD research.

I don't see any problem if the information is shared provided the results will be disseminated…back to the people who volunteered their information to be shared. (FG6 Dar es Salaam)

… if it is going to help others, then I don't have any problem about it. (FG2 Kumasi)

I agree only if the data will be shared to advance discovery only in sickle cell disease in other words… don't want to share their data with heart disease. (FG4 Kano)

Some participants interrogated the notion of data sharing broadly and raised specific concerns, for example, in relation to use of information by healthcare insurance companies and for supernatural practices.

I just want to know who the group will be sharing the information with because sometimes information can be used against the participant. I am talking about the group, the insurance comes to mind now. (FG5 Lagos)

I will have no concern about sharing the data as long as it is not going to witchcraft people. (FG3 Abuja)

### Duty of care

The most contentious issue debated was about healthcare provision for people with SCD. A unique opportunity to confront healthcare policy-makers led to passionate arguments that revealed numerous difficulties consistent at all the study sites. Summary findings are presented in

table 4. A full account of this interaction will be reported in a separate publication.

### DISCUSSION

Community engagement is a key strategy for improving outcomes in research by proactively seeking out values, concerns and aspirations within communities. This is a core component of SickleGenAfrica, and we identified important ethical issues from our initial focus groups.

First, examination of autonomy in research participation decision-making revealed that a minority of participants identified with community decision-making and a need to consult or seek approval from significant others, including household heads, family members, and religious and community leaders. Nevertheless, participants who indicated autonomous decision-making would also inform significant others of their decision. For example, parents/caregivers (mostly mothers) who are primarily and legally responsible for a child with SCD under the age of 16 years also believed they could make informed research participation decisions independently. However, they would inform the child's father subsequently. There are no legal requirements for both parents to give consent in the three countries. Decision-making is a process of identifying and choosing alternatives based on values, preferences and beliefs. In Africa, it has been suggested that the Ubuntu philosophy ('I am because we are') of shared decision-making, for example, may be more appropriate than self-determination due to family and communal systems[7] and may ensure encouragement in research participation. Nonetheless, consistent with our findings, a recent study of shared decision-making within the context of consent for genomics research did not support the Ubuntu philosophy; shared decision-making was considered to be important in specific cases such as with vulnerable people,[8] for example, parents/caregivers of children with SCD, who are stigmatised and rely on family or community for their support.

Second, some sensitivity was raised about biobanking and genomics data sharing. Although the majority of participants were not concerned about these components of SickleGenAfrica, some issues were articulated reflecting societal realities and cultural perspectives. For example, constant power outage in Nigeria was highlighted as a systemic problem that may affect the preservation of samples in a biorepository together with identification, misuse or loss of samples and data. Biorepositories hold biological samples linked to databases and could be at risk of breaching privacy.[9] Protecting biorepository samples and the identity of participants is a fundamental ethical issue; however, there is not a uniform governance framework for genomic biobanking in Africa.[10] Therefore, participants were assured that researchers have to strictly ensure trustworthiness, integrity and competence in the storage of biological samples. They were informed the biorepository had adequate backup power and has been used to store samples from other H3Africa projects

**Table 4** SickleGenAfrica year 1 focus groups: duty of care themes and sample participant vignettes

| Themes | Vignettes |
| --- | --- |
| Medical care and health insurance | "The other thing is health insurance as Arafa had been working on, you need to be in a group and they don't want sickle cell patients as they won't get profit. This one has been hurting us because I cannot pay 1.5 million shillings for my NHIF insurance, I don't have that money." (FG6 Dar es Salaam)<br>"…the mother has to pay and an average stay in Korle-Bu without insurance a day is about 350 Cedis, where is the mother going to get the money from. So facilities, subventions on their drugs and then education is very, very important." (FG1 Accra) |
| Improving quality of care | "Okay, what I also want the government to do is that like we have in other countries and even some in Africa, they have comprehensive sickle cell care…." (FG2 Kumasi)<br>"There should be a guideline on what should be done depending on level of health facility. So it's best for each level to know what it can do and what they can't so as to know when to refer." (FG6 Dar es Salaam) |
| Inadequate resources and infrastructure | "Government should try and provide some specific drugs for us… because most of our drugs are very expensive… and to even get it is really difficult…" (FG5 Lagos)<br>"Government is to come to our aid and they should provide us with the facilities and infrastructure because Korle-Bu sickle cell unit don't run 24 hours." (FG1 Accra)<br>"All sickle cell patients in Murtala do have an ID card so with their ID card whenever they are in crisis and they happen to be in the A&E they don't wait, they don't follow any queue and then they have the desired attention for them to get cured of that crisis." (FG4 Kano) |
| Inadequate qualified staff | "We have clinics and then we have just general practitioners running the clinics, but what we have done in the State now is we employ haematologists, we have about 12, 13 centres in the state where you can access these haematologists…." (FG5 Lagos)<br>"The issue of health provider is still a big challenge as they are not enough so you might find may be one person is running the clinic and there are a lot of patients." (FG6 Dar es Salaam)<br>"The evidence is that because of lack of healthcare, by five years most of this children have died and if you go to our hospitals in the rural area you will find communities where there is no health facilities." (FG4 Kano) |
| Enhancing human resources | "Sickle cell disease experts should be increased as you might find some doctors attending not know much about the disease, which might compromise care given." (FG6 Dar es Salaam)<br>"I want government to train doctors and nurses and those working in the hospital so that they can set up sickle cell clinics or hospitals so the sickle cell patients wouldn't have to travel long distances to access healthcare." (FG2 Kumasi) |
| Poor healthcare provider behaviour | "…once you get to the hospital as my fellows have said, you give explanation to the doctors that one two and three and he replied no, I am the doctor and you are a patient. Don't teach me how to do my work." (FG6 Dar es Salaam)<br>"…my child for instance they gave her 6 months' time to visit but within a month she had crisis, and if you bring the child, they will not even pick your file for you, they will just ask you to take your child away, where should we take them to?" (FG2 Kumasi) |
| Socioeconomic concerns of parents and caregivers | "Apart from the four children, I live with my mother after my father died all of them are my responsibility. So if today you say there is something needed it means other children should wait." (FG6 Dar es Salaam)<br>"There was a time they were turning down patients that are SS and don't have money, they will tell them there is no bed." (FG5 Lagos)<br>"…sometimes is very difficult to make the payment or buying of the drugs and this thing has compounded their sickness." (FG1 Accra) |
| Perceived stigma and misconceptions | "…Just put some funds into it, let there be jingles, let there be information, let people just know so that this stigmatisation can stop." (FG5 Lagos)<br>"…on the issue of stigmatization, it is something which is going on especially with some of the health workers, sometimes you go and they will see SS as if your child will die in the next moment, so sometimes I was telling a doctor that stigmatisation starts from the hospital." (FG2 Kumasi)<br>"There are still many people who don't know about the disease with others feeling like they will be dying once diagnosed." (FG6 Dar es Salaam) |

**Table 4** Continued

| Themes | Vignettes |
|---|---|
| Social neglect and isolation | "I am desperate for a cure. I can't tell you so many thing my mum tried, my own dad, my story is a bit different because my own dad neglected me so my mum was trying everything, both traditional, both orthodox." (FG3 Abuja)<br>"…some parents if they have sickle cell disease patients and children and they don't want to buy drugs for them because they believe they will not survive…." (FG4 Kano)<br>"You find out that some men they run away. They leave their children to the mothers to take care of, once the crisis sets in and they feel that I can't stand it any longer and so they leave the mother to take care of these babies. It is so demoralizing when you hear this and see it being practiced." (FG5 Lagos) |
| Laws and policies | "I have some patients who say they do not want disclosure to be made to their employers regarding their sickle cell status because they would be fired or lose their jobs it is very important, I think there is a law that we should not discriminate….so I think it is very very important that is enforced." (FG1 Accra)<br>"Kano State Government should adopt something like what is called Medicare in the United States. It should specifically make anybody with sickle cell disease….under the Medicare should have complete coverage." (FG4 Kano) |
| Private and public sector partnerships | "We have organizations, churches, NGOs can decide to sponsor people and say okay I can afford to pay for 20 people for a whole year, I can afford to pay for ten families or an organization can come and say okay…lets pay for 50 000 people for the whole year." (FG5 Lagos)<br>"…also looking at corporate bodies partnership, strategic partnership with cooperate bodies you know there are certain organizations that are already supporting other health issues…but we don't have any strong cooperate partnership with say MTN or Vodafone, so that we can be able to harness resource it could be technical, it could be financial…" (FG1 Accra) |
| Public awareness | "I want the government to take up the costs of creating awareness so that the government ensures that every district directorate or health directorate should try and educate the general public on the sickle cell disease." (FG2 Kumasi)<br>"…she is calling on government to increase awareness in the rural communities." (FG4 Kano)<br>"….so as to create awareness to the patients especially in the villages where some lives are lost without knowing and nobody asks." (FG6 Dar es Salaam) |

SickleGenAfrica, Sickle Cell Disease Genomics of Africa Network.

for several years. Additionally, participants were informed that ensuing community engagement workshops will include video presentations to illustrate SickleGenAfrica research processes, including enrolment, sample storage in the biorepository (showing alternative power supply) and de-identification for data sharing.

Third, a majority of participants supported the return of individual genomic findings with a beneficial viewpoint regardless of treatment unavailability at the time, but with appropriate counselling and psychosocial support. For some participants, however, they preferred not to be informed since this could generate anxiety and prejudices against those with SCD. Our results are consistent with previous research showing increasing expectation in genomics for reporting discoveries of clinical significance with some recommendations.[11 12] Our SickleGenAfrica informed consent document records participants' preferences about receiving feedback of genomic findings and broad consent for future research. We are collecting additional information from research participants in a survey. Consideration has to be made about returning individual findings, including availability of interventions and resources for conveying information.[13] We

impressed upon the participants that return of individual results in SickleGenAfrica was not standard procedure; however, this will be contemplated for both predicted and incidental findings within the framework of H3Africa ethical guidelines and international consensus. Furthermore, genetic counselling, psychosocial support and specialist SCD resources including clinics need to be in place. SickleGenAfrica is collaborating with the West African Genetic Medicine Centre (University of Ghana) to train genetic counsellors who could be deployed for this purpose. Current international deliberations aim to address two areas of importance: individual return of results and genomic summary results (GSR) for data sharing. A recent review of country laws and policies concerning the return of results categorised these as the following: must return, should return, may return or do not return, with no harmonisation.[14] Moreover, there are insufficient guidelines on returning results to families of deceased people.[15] On the other hand, a 2018 amendment to the National Institutes of Health policy demands GSR to be openly available online without regulation.[16] This may expose vulnerable populations in Africa and is being debated within H3Africa. In both areas, our

community engagement work will contribute to ongoing discussions. KAA is a member of the H3Africa Ethics and Community Engagement Working Group. We intend to include GSR during our second round of community engagement activities.

Fourth, closely linked to returning genomic findings is duty of care. Across the study sites, participants expected governments principally to prioritise SCD and provide adequate care. Highlighted areas included well-resourced SCD centres with specialists, 24-hour services in general hospitals and local clinics, subsidised or free treatment, diagnostic laboratories, staff training, private sector involvement, and public education to curtail stigmatisation and discrimination. Although there were some assurances from healthcare policy-makers that the healthcare-related issues raised were being addressed, participants disagreed. In our experience, patients with SCD and their families in Africa tend to raise issues about the lack of access to adequate care when they are engaged in research. Therefore, this was an opportunity to involve policy-makers at the outset of research that will potentially have no benefits for participants.

There were some limitations to our study. First, we identified a potential bias in the selection of participants by invitation to the focus groups, and the targeted number of participants per site (n=20) was not entirely achieved. Second, lay people and those with low literacy levels were expected to provide comments on community concerns (both ethical and social) in genomics research. Although we provided lay explanations with demonstrations and translators, their comprehension may have been limited or related to conventional research with minimal understanding of the specific ethical issues associated with genomics research. Third, involvement of community leaders in community engagement activities concerning stigmatised populations, including SCD, may breach confidentiality.[17] However, their experience of handling sensitive and confidential issues may help protect vulnerable people from risks and harm. Fourth, some parents/caregivers attended the focus groups with their children who were not active contributors to the discussions. Future community engagement will actively involve children.

In conclusion, we achieved our goal of informing participants about SickleGenAfrica genomics research before recruitment and gathered important insights and sensitivities concerning ethical and societal implications of this research within different cultural settings. Our results together with other H3Africa research will contribute specifically to the ongoing appraisal of H3Africa ethics and community engagement policies and other international guidelines for genomics research broadly.

**Author affiliations**
[1]Faculty of Medicine, Imperial College London, London, UK
[2]Haematology and Sickle Cell Centre, London North West University Healthcare NHS Trust, London, UK
[3]Department of Haematology, University of Ghana Medical School, University of Ghana, Accra, Ghana
[4]Ghana Institute of Clinical Genetics, Accra, Ghana
[5]School of Medical Sciences, Kwame Nkrumah University of Science and Technology, Kumasi, Ghana
[6]Directorate of Child Health, Komfo Anokye Teaching Hospital, Kumasi, Ghana
[7]School of Public Health, Kwame Nkrumah University of Science and Technology, Kumasi, Ghana
[8]Department of Haematology and Blood Transfusion, College of Medicine, University of Lagos, Lagos, Nigeria
[9]Department of Medicine, Bayero University and Aminu Kano Teaching Hospital, Kano, Nigeria
[10]Department of Epidemiology, School of Public Health, University of Alabama at Birmingham, Birmingham, Alabama, USA
[11]Department of Haematology and Centre of Excellence for Sickle Cell Disease Research and Training, University of Abuja, Abuja, Nigeria
[12]Sydney Brenner Institute for Molecular Bioscience, University of the Witwatersrand, Johannesburg, South Africa
[13]School of Biomedical and Allied Health Sciences, College of Health Sciences, University of Ghana, Legon, Ghana
[14]Community Liaison, Accra, Ghana
[15]Community Liaison, Kumasi, Ghana
[16]Community Liaison, Lagos, Nigeria
[17]Community Liaison, Kano, Nigeria
[18]Community Liaison, Abuja, Nigeria
[19]Community Liaison, Dar es Salaam, Tanzania, United Republic of
[20]West African Genetic Medicine Centre (WAGMC), College of Health Sciences, University of Ghana, Legon, Ghana

**Acknowledgements** We are extremely grateful to all the participants of the Community Engagement Focus Groups at the six SickleGenAfrica study sites and Mary Akua Ampomah (MAA) for her role as the second psychologist.

**Collaborators** SickleGenAfrica Network Members: (1). Coordinating Centre: Amma Benneh-Akwasi Kuma, Anita Ghansah, Catherine Segbefia, David Nana Adjei, Edeghonghon Olayemi, Gordon Awandare, Solomon F. Ofori-Acquah, William Kudzi (Korle Bu Teaching Hospital, Accra, Ghana and University of Ghana, Accra, Ghana). (2). Collaborative Sites: Ellis Owusu-Dabo, Vivian Painstil (Komfo Anokye Teaching Hospital, Kumasi, Ghana and Kwame Nkrumah University of Science and Technology, Kumasi, Ghana); Aisha Kuliya-Gwarzo, Adullahi Shehu, Baba Musa, Mahmoud Sani, Najibah Aliyu Galandanci (Aminu Kano Teaching Hospital, Kano, Nigeria, Bayero University, Kano, Nigeria and Murtala; Mohammad Specialist Hospital, Kano, Nigeria); Alashle Abimiku, Ameh Adeyefa, Obiageli E. Nnodu (Institute of Human Virology Nigeria, Abuja, Nigeria and University of Abuja, Abuja, Nigeria); Michael Akinsete, Olufunto Kalejaiye, Titilope Adeyemo (Lagos University Teaching Hospital, Lagos, Nigeria and University of Lagos, Nigeria); Furahini Tluway, Flora Ndobho, Josephine Mgaya, Julie Makani, Siana Nkya (Muhimbili University of Health and Allied Sciences, Dar es Salaam, Tanzania); Nicola Mulder (University of Cape Town, Cape Town, South Africa); Kofi A Anie MBE (Imperial College London, London, UK); Jonathan Stiles (Morehouse School of Medicine, Atlanta, USA); Flordeliza Villanueva, Samit Ghosh, Solomon Ofori-Acquah, Ryan Minster (University of Pittsburgh, Pittsburgh, USA).

**Contributors** KAA led the conception, design and development of the community engagement effort with substantial contributions from EO, VP, EO-D, TAA, MUS, NAG, ON, FT, DNA, JO-L and SO-A, and community liaisons PM, JS-A, HN, AG, AB and AS. All authors are accountable for specific aspects of the work.

**Funding** SickleGenAfrica is carried out as a collaborative project funded by the National Heart, Lung, and Blood Institute, USA (grant number 1U54HL141011-01).

**Competing interests** None declared.

**Patient consent for publication** Not required.

**Ethics approval** Research ethics approvals were obtained from all six institutions where patient recruitment took place (table 1).

**Provenance and peer review** Not commissioned; externally peer reviewed.

**Data availability statement** All data relevant to the study are included in the article or uploaded as supplementary information.

**ORCID iD**
Kofi A Anie http://orcid.org/0000-0003-0513-3331

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
