## [Reviewer comments · BMJ Open]

ARTICLE DETAILS

TITLE (PROVISIONAL)	Sickle Cell Disease Genomics of Africa (SickleGenAfrica) Network: Ethical Framework and Initial Qualitative Findings from Community Engagement in Ghana, Nigeria, and Tanzania
AUTHORS	Anie, Kofi; Olayemi, Edeghonghon; Paintsil, Vivian; Owusu-Dabo, Ellis; Adeyemo, Titilope; Sani, Mahmoud; Galadanci, Najibah; Nnodu, Obiageli; Tluway, Furahini; Adjei, David; Mensah, Peter; Sarfo-Antwi, Joseph; Nwokobia, Henry; Gambo, Awwal; Benjamin, Adebola; Salim, Arafa; Osae-Larbi, Judith; Ofori-Acquah, Solomon

VERSION 1 – REVIEW

REVIEWER	Nicol, Dianne University of Tasmania, Law
REVIEW RETURNED	14-Feb-2021

GENERAL COMMENTS	I note that I am reviewing this manuscript for the second time. For some reason, I have not been able to view a version with track changes so that I can readily see how the authors have responded to the feedback provided by the three reviewers. However, I have read their response letter to the editor and the clean version of the manuscript. Based on this, I am content that they have responded to feedback and recommend publication.
--

REVIEWER	Vinjamur, Divya Boston Children's Hospital, Hematology and Oncology
REVIEW RETURNED	22-Feb-2021

GENERAL COMMENTS	In this revision, Kofi A, et al., have addressed all my previous comments. I am satisfied with the revised manuscript. See below a few minor comments to improve the readability of the manuscript. Page 7 line 5: Is the second psychologist not referred to by her initials because she is not an author on this study? Both psychologists should be referred to in a similar manner, i.e. initials for both or "...focus groups were moderated by two psychologists, both experienced in SCD.." Page 18 line 4: Do the authors mean to say that there will be potential benefits for participants? If so, remove "no" from this sentence. Page 18 last sentence: "Our results together with will contribute.." - together with what? Please check for missing words. The Discussion section requires proofreading before publication.
---

	The “Best Interest” statement in Table 3 is not clear. Please expand/clarify.
--	---

VERSION 1 – AUTHOR RESPONSE

Reviewers' Comments to Author:

Reviewer: 2

Page 7 line 5: Is the second psychologist not referred to by her initials because she is not an author on this study? Both psychologists should be referred to in a similar manner, i.e. initials for both or “...focus groups were moderated by two psychologists, both experienced in SCD..”

This has been corrected. Initials for both psychologists are included.

Page 18 line 4: Do the authors mean to say that there will be potential benefits for participants? If so, remove “no” from this sentence.

No, consistent with genomic research in general, this research will potentially have no benefits to the participants.

Page 18 last sentence: “Our results together with will contribute..” - together with what? Please check for missing words.

This has been checked and amended, thank you.

The Discussion section requires proofreading before publication.

This has been done. The entire manuscript will be proof read before publication.

The “Best Interest” statement in Table 3 is not clear. Please expand/clarify.

This has been clarified.

VERSION 2 – REVIEW

REVIEWER	Vinjamur, Divya Boston Children's Hospital, Hematology and Oncology
REVIEW RETURNED	28-Jun-2021

GENERAL COMMENTS	Dear Authors, All my comments have been addressed and I am satisfied with the revised manuscript. Please proofread the Discussion section for spelling errors before publication. Best wishes.
--